# Janus-Type AIE Fluorophores: Synthesis and Properties of π-Extended Coumarin-Bearing Triskelions

**DOI:** 10.3390/molecules27217450

**Published:** 2022-11-02

**Authors:** Masafumi Ueda, Mirai Kokubun, Nao Yanagi, Norifumi Yamamoto, Yasuhiro Mazaki

**Affiliations:** 1Department of Chemistry, Graduate School of Science, Kitasato University, Kanagawa 252-0373, Japan; 2Department of Applied Chemistry, Faculty of Engineering, Chiba Institute of Technology, Chiba 275-0016, Japan

**Keywords:** aggregation-induced emission enhancement, distorted π-fluorophore, π-extended coumarin, helical inversion, Janus-type triskelion

## Abstract

Janus-type triskelion-shaped fluorophores comprising coumarins bearing various electron-donating substituents (**1aad**, **1add**, **1ccd**, and **1cdd**) were successfully synthesized via an intramolecular Ullmann coupling. Density functional theory (DFT) calculations indicated that all the compounds presented two different molecular surfaces, similar to Janus-type molecules. The absorption and fluorescence spectra of asymmetrical derivatives **1aad**, **1add**, **1ccd**, and **1cdd** exhibited a bathochromic shift due to their narrow highest occupied molecular orbital (HOMO) –lowest unoccupied molecular orbital (LUMO) gap. Natural transition orbital (NTO) analysis indicated that the excited state orbital overlaps differ among the *C*_3_ symmetrical and asymmetrical dyes. These triskelion-shaped fluorophores were found to form molecular nanoaggregates in THF/H_2_O mixtures and demonstrated aggregation-induced emission (AIE) enhancement characteristics as a result of restricting their molecular inversion. These results indicate that Janus-type AIE fluorophores are potentially applicable as solid-state fluorescent chiral materials, which can be optimized by controlling their molecular rearrangement in the solid state.

## 1. Introduction

Aggregation-induced emission (AIE) is an important characteristic of some molecular luminescent materials, such as 1,1,2,3,4,5-hexaphenylsilole (HPS) [1] and tetraphenylethene (TPE) [2]. These derivatives and their analogs have been extensively applied in bio-imaging, chemical sensing, organic light-emitting diodes (OLEDs), and circularly polarized luminescence (CPL) because they exhibit strong emission upon the formation of nanoaggregates under appropriate concentration conditions [3]. Nanoaggregated molecules can be in close proximity to each other while maintaining an appropriate spatial distance. Therefore, this unusual behavior is elicited by restricting molecular rotations, vibrations, and motions, which are the major causes of non-radiative decay. Conversely, reducing intermolecular interactions, such as π···π interactions in nanoaggregates, is among the strategies for enhancing luminous efficiency. For instance, it is possible to mitigate intermolecular forces by introducing bulky substituents into the target molecules. This chemical modification is also efficient in suppressing the rotation of HPS and TPE [4,5]. Furthermore, 1,4,5,8,9,12-hexamethyltriphenylene (HMTP, Figure 1a) adopts a twisted structure as a result of intramolecular steric repulsion of the adjacent methyl groups [6]. This twisting of HMTP inhibits intermolecular π-stacking, resulting in highly efficient emission in the solid state [7]. Thus, molecular twisting plays a vital role in controlling molecular association. We focused on the enhanced emission intensity caused by molecular twisting and hypothesized that this phenomenon applies to distorted π-conjugated molecules. Based on these insights and ideas, we initiated the development of distorted fluorophores as components of novel AIE luminogens (AIE-gens).

The molecular framework of 1,3,5-triphenylbenzene (TPB) is suitable for this purpose and has often been used as a building block in optical devices [8]. Furthermore, bridging between the central benzene ring and the outer phenyl groups enhances rigidity, bending, and symmetry. TPB is bridged via a sterically crowded lactone chain to form the triskelion-shaped molecules bearing coumarin units (**1**). This triskelion-shaped framework is expected to adopt a distorted geometry owing to the intramolecular repulsion between the oxygen atoms of the carbonyl groups and the hydrogen atoms of the outer phenyl units.

Coumarins with electron-donating groups at the 7-position exhibit strong fluorescence emission due to intramolecular charge transfer (ICT, Figure 1b) [9,10]. Their photophysical properties are promising for laser dyes [9,10,11], molecular sensors [12,13,14], and solar cells [15]. Furthermore, numerous extended coumarin dyes have been synthesized, and the photophysical properties based on their rigid and planar structures have been described [16,17,18,19,20,21,22,23,24]. Conversely, unusual coumarin dyes with helical geometries have also been reported; these compounds represented a platform for the development of molecular chiral dopants [25,26,27]. Thus, the design of novel coumarin dyes utilizing both their emission characteristics and topologies is required for advancing next-generation optical materials.

Our group has previously succeeded in synthesizing *C*_3_ symmetric triskelion-shaped coumarin dye (**1A**) and its donor-substituted derivatives (**1B**–**1E**, Figure 1c). Their molecular structures and photophysical properties have been reported [28,29]. X-ray crystallography revealed that the molecular framework of these compounds presented a helical geometry arising from intramolecular steric repulsion. Furthermore, in the crystal structures of **1A**–**1D**, the left-handed (*M,M,M*-**1**) and right-handed (*P,P,P*-**1**) structural isomers were presented in a 1:1 ratio. Theoretical calculation indicated that these molecules rapidly underwent racemization due to a low helical inversion barrier. Interestingly, these distorted coumarin dyes exhibited AIE characteristics owing to the formation of the nanoaggregates in the THF/H_2_O mixture. That is, we achieved control of conformational changes in the internal conversion process by aggregating twisted molecules. Thus, we have found the potential of this coumarin-based framework as AIE fluorophores.

As compound **1** possesses three coumarin units, different electron-donating groups can be introduced synthetically into the coumarin units. Such a chemical modification with various combinations of the substituents is effective with respect to tuning molecular orbital levels and introducing asymmetry, and it is an interesting effect on the luminescence property. There, we designed novel asymmetric triskelion-shaped coumarin dyes with several donor groups (**1aad**, **1add**, **1ccd**, and **1cdd**, Figure 1d) and investigated their electronic structures and photophysical properties based on the asymmetric distorted structures. Herein, we describe the synthesis and AIE properties of asymmetrical triskelion-shaped fluorophores **1aad**, **1add**, **1ccd**, and **1cdd**. To better understand these coumarin-bearing triskelions, they were compared with *C*_3_ symmetric triskelions **1A**–**1E**.

## 2. Materials and Methods

All reagents were commercially sourced. Mesitylene, iron powder, bromine, *N*-bromosuccinimide, benzoyl peroxide, sodium acetate, thionyl chloride, potassium carbonate, 1-bromo-3-chloropropane, *m*-aminophenol, 1,5-dibromopentane, *N*-ethyldiisopropylamine, copper powder, and organic solvents were purchased from FUJIFILM Wako pure Chemical Co., Ltd. (Osaka, Japan). 2-Bromophenol and 3-methoxyphenol were purchased from Tokyo Chemical Industries Co., Ltd. (Tokyo, Japan). Melting points were determined using a Yanaco MP-500P micro-melting point apparatus. ^1^H (600 or 400 MHz) and ^13^C (150 or 100 MHz) nuclear magnetic resonance (NMR) spectra were recorded using a Bruker AVANCE instrument. The following abbreviations were used to describe the multiplicities: singlet (s), doublet (d), triplet (t), doublet of doublets (dd), and multiplet (m). Absorption spectra were recorded using a JASCO V-560 instrument. High-resolution mass spectrometry (HRMS) spectra were recorded on a Thermo Fisher Scientific (Tokyo, Japan), Exactive Plus Orbitrap mass spectrometer for ionization. Only relatively intense peaks and structurally diagnostic mass spectral fragment ion peaks are reported. All quantum chemical calculations were performed using the Gaussian 16 program (Revision C.01) [30]. The optimized structures of symmetrical and asymmetrical triskelion-shaped molecules were determined using density functional theory (DFT) calculations. All DFT calculations were performed using the ωB97-XD long-range corrected hybrid functional and 6-31G(d) basis set. Geometry optimizations were performed for the symmetrical derivatives while maintaining the *C*_3_ symmetry. Normal mode analysis calculations were conducted at the same level of theory to ensure that local minima optimized structures were obtained. Electronic excitation energies and natural transition orbitals [31] were estimated for all triskelion-shaped derivatives using time-dependent (TD)-DFT calculations. Fluorescence spectra were collected using a HITACHI F-4500 fluorescence spectrometer and a JASCO FP-8550 spectrofluorometer. Relative fluorescence quantum yields (*F*_F_) were determined using rhodamine B in EtOH as the standard (*F*_F_ = 0.9). Dynamic light scattering (DLS) measurements were recorded using a Sysmex Zetasizer NanoZS instrument.

The syntheses of **1A**–**E** and **8A**–**E** have been described in our previous reports [28,29].

**Synthesis of 9.** To a 50 mL recovery flask equipped with a reflux condenser were added K_2_CO_3_ (28 mg, 0.20 mmol), **7d** (50 mg, 0.20 mmol), **6** (102 mg, 0.20 mmol), and acetone (15 mL). The reaction mixture was refluxed overnight. After cooling to ambient temperature, residual solids were removed by filtration and washed with CHCl_3_. The filtrate was then evaporated under reduced pressure. The residue was purified by column chromatography on silica gel (CH_2_Cl_2_: hexane = 4:1 (*v*/*v*)) to obtain **9** (81 mg, 0.11 mmol) as yellow block crystals in 17% yield. Mp = 155 °C (decomp.). ^1^H NMR (400 MHz, CDCl_3_) δ 7.44 (d, *J* = 8.8 Hz, 1H), 6.88 (d, *J* = 2.8 Hz, 1H), 6.76 (dd, *J* = 2.8 and 8.8 Hz, 1H), 3.17 (t, *J* = 5.6 Hz, 4H), 1.70 (m, 4H), 1.62–1.58 (m, 2H). ^13^C NMR (100 MHz, CDCl_3_) δ 165.2, 161.5, 152.5, 147.9, 141.8, 138.2, 133.7, 117.6, 116.0, 114.7, 110.2, 102.6, 50.0, 25.6, 24.2. UV-vis (CH_2_Cl_2_, *c* = 1.0 × 10^−5^ M) λ_max_ (*ε*) 261 (17,300) nm. HRMS (ESI, positive mode): *m*/*z* calcd for C_20_H_14_Br_4_Cl_2_NO_4_: [M + H]^+^ 721.6987; found: 721.6986.

**Synthesis of 10.** To a 50 mL recovery flask equipped with a reflux condenser were added K_2_CO_3_ (83 mg, 0.60 mmol), **7d** (152 mg, 0.60 mmol), **6** (150 mg, 0.30 mmol), and acetone (15 mL). The reaction mixture was refluxed overnight. After cooling to ambient temperature, the residual solids were removed by filtration and washed with CHCl_3_. The filtrate was then evaporated under reduced pressure and the residue was purified by column chromatography on silica gel (CH_2_Cl_2_: hexane = 4:1 (*v*/*v*)) to obtain **10** (140 mg, 0.15 mmol) as yellow block crystals in 50% yield. Mp = 185 °C (decomp.). ^1^H NMR (400 MHz, CDCl_3_) δ 7.45–7.43 (m, 2H), 6.92–6.91 (m, 2H), 6.77–6.74 (m, 2H), 3.19–3.16 (m, 8H), 1.73–1.68 (m, 8H), 1.62–1.58 (m, 4H). ^13^C NMR (100 MHz, CDCl_3_) δ 165.5, 161.9, 152.5, 147.9, 141.7, 138.0, 133.7, 120.0, 117.1, 116.0, 110.3, 102.8, 50.1, 25.6, 24.2. UV-vis (CH_2_Cl_2_, *c* = 1.0 × 10^–5^ M) λ_max_ (*ε*) 316 (4400), 263 (34,100) nm. HRMS (ESI, positive mode): *m*/*z* calcd for C_31_H_26_Br_5_ClN_2_O_5_: [M + H]^+^ 942.7459; found: 942.7455.

**Synthesis of 8aad.** To a 50 mL recovery flask equipped with a reflux condenser were added K_2_CO_3_ (39 mg, 0.28 mmol), **7a** (30 μL, 0.28 mmol), **9** (100 mg, 0.14 mmol), and acetone (15 mL). The reaction mixture was refluxed overnight. After cooling to ambient temperature, the residual solids were removed by filtration and washed with CHCl_3_. The filtrate was then evaporated under reduced pressure. The residue was purified by column chromatography on silica gel (CH_2_Cl_2_: hexane = 4:1 (*v*/*v*)) to obtain **8aad** (103 mg, 0.10 mmol) as yellow block crystals in 74% yield. Mp = 98 °C (decomp.). ^1^H NMR (400 MHz, CDCl_3_) δ 7.69 (dd, *J* = 1.6 and 8.0 Hz, 2H), 7.50 (dd, *J* = 1.6 and 8.0 Hz, 2H), 7.45 (d, *J* = 8.8 Hz, 2H), 7.41 (dd, *J* = 1.6 and 8.0 Hz, 1H), 3.18 (t, *J* = 5.6 Hz, 4H), 1.73–1.68 (m, 4H), 1.62–1.58 (m,2H). ^13^C NMR (100 MHz, CDCl_3_) δ 162.2, 162.1, 152.5, 148.0, 147.6, 137.9, 137.7, 134.1, 133.7, 128.7, 128.3, 123.5, 119.6, 119.5, 116.0, 115.6, 110.4, 102.8, 50.1, 25.6, 24.2. UV-vis (CH_2_Cl_2_, *c* = 1.0 × 10^–5^ M) λ_max_ (*ε*) 266 (15,500) nm. HRMS (ESI, positive mode): *m*/*z* calcd for C_32_H_21_Br_6_NO_6_: [M + H]^+^ 995.6481; found: 995.6480.

**Synthesis of 8add.** To a 50 mL recovery flask equipped with a reflux condenser were added K_2_CO_3_ (15 mg, 0.11 mmol), **7a** (11 μL, 0.11 mmol), **10** (101 mg, 0.11 mmol), and acetone (15 mL). The reaction mixture was refluxed overnight. After cooling to ambient temperature, the residual solids were removed by filtration and washed with CHCl_3_. The filtrate was then evaporated under reduced pressure. The residue was purified by column chromatography on silica gel (CH_2_Cl_2_: hexane = 4:1 (*v*/*v*)) to obtain **8add** (66 mg, 0.06 mmol) as colorless block crystals in 58% yield. Mp = 268 °C (decomp.). ^1^H NMR (400 MHz, CDCl_3_) δ 7.70–7.68 (m, 1H), 7.52–7.49 (m, 1H), 7.46–7.45 (m, 2H), 7.44–7.40 (m, 1H), 7.24–7.20 (m, 1H), 6.96–6.95 (m, 1H), 6.77–6.74 (m, 2H), 3.20–3.17 (m, 8H), 1.74–1.69 (m, 8H), 1.63–1.59 (m, 4H). ^13^C NMR (100 MHz, CDCl_3_) δ 162.3, 162.2, 152.5, 148.1, 147.7, 138.0, 134.2, 133.7, 128.7, 128.3, 123.5, 119.7, 119.5, 116.0, 115.6, 110.5, 102.9, 50.2, 25.7, 24.2. UV-vis (CH_2_Cl_2_, *c* = 1.0 × 10^–5^ M) λ_max_ (*ε*) 264 (31,700) nm. HRMS (ESI, positive mode): *m*/*z* calcd for C_37_H_30_Br_6_N_2_O_6_: [M + H]^+^ 1078.7216; found: 1078.7217.

**Synthesis of 8ccd.** To a 50 mL recovery flask equipped with a reflux condenser were added K_2_CO_3_ (40 mg, 0.28 mmol), **7c** (70 mg, 0.28 mmol), **9** (101 mg, 0.14 mmol), and acetone (15 mL). The reaction mixture was refluxed overnight. After cooling to ambient temperature, the residual solids were removed by filtration and washed with CHCl_3_. The filtrate was then evaporated under reduced pressure. The residue was purified by column chromatography on silica gel (CH_2_Cl_2_) to obtain **8ccd** (78 mg, 0.07 mmol) as yellow block crystals in 52% yield. Mp = 197 °C (decomp.). ^1^H NMR (400 MHz, CDCl_3_) δ 7.45–7.42 (m, 3H), 6.96–6.95 (m, 1H), 6.76–6.73 (m, 3H), 6.55–6.52 (m, 2H), 3.19–3.16 (m, 4H), 2.97 (s, 12H), 1.73–1.68 (m, 4H), 1.62–1.58 (m, 2H). ^13^C NMR (100 MHz, CDCl_3_) δ 162.3, 162.3, 152.5, 150.8, 148.1, 137.9, 137.9, 133.6, 119.5, 119.5, 115.9, 112.4, 110.5, 106.7, 102.9, 100.4, 50.1, 40.6, 25.7, 24.2. UV-vis (CH_2_Cl_2_, *c* = 1.2 × 10^–5^ M) λ_max_ (*ε*) 311 (7400), 266 (38,300), 261 (38,100), 255 (31,400) nm. HRMS (ESI, positive mode): *m*/*z* calcd for C_36_H_32_Br_6_N_3_O_6_: [M + H]^+^ 1081.7325; found: 1078.7327.

**Synthesis of 8cdd.** To a 50 mL recovery flask equipped with a reflux condenser were added K_2_CO_3_ (20 mg, 0.14 mmol), **7c** (30 mg, 0.14 mmol), **10** (131 mg, 0.14 mmol), and acetone (20 mL). The reaction mixture was refluxed overnight. After cooling to ambient temperature, the residual solids were removed by filtration and washed with CHCl_3_. The filtrate was then evaporated under reduced pressure. The residue was purified by column chromatography on silica gel (CH_2_Cl_2_) to obtain **8cdd** (78 mg, 0.07 mmol) as white block crystals in 67% yield. Mp = 197 °C (decomp.). ^1^H NMR (400 MHz, CDCl_3_) δ 7.46–7.42 (m, 3H), 6.96–6.95 (m, 2H), 6.77–6.73 (m, 3H), 6.55–6.53 (m, 1H), 3.19–3.17 (m, 8H), 2.97 (s, 6H), 1.74–1.68 (m, 8H), 1.62–1.58 (m, 4H). ^13^C NMR (100 MHz, CDCl_3_) δ 162.3, 162.3, 152.5, 150.8, 148.1, 148.1, 137.9, 137.9, 133.7, 119.5, 119.5, 115.9, 112.4, 110.5, 106.7, 102.9, 100.4, 50.1, 40.6, 25.7, 24.4. UV-vis (CH_2_Cl_2_, *c* = 1.0 × 10^–5^ M) λ_max_ (*ε*) 319 (7300), 266 (48,100), 259 (42,300), 254 (28,600). HRMS (ESI, positive mode): *m*/*z* calcd for C_39_H_36_Br_6_N_3_O_6_: [M + H]^+^ 1121.7638; found: 1121.7638.

**Synthesis of 1aad.** To a 50 mL Schlenk tube was added **8aad** (100 mg, 0.10 mmol), excess activated Cu Powder (1410 mg, 22.0 mmol), and dry DMF (30 mL) under argon atmosphere. The reaction mixture was refluxed for 18 h. After cooling to ambient temperature, the reaction mixture was filtered through Celite and washed with EtOAc. The organic phase was then washed with H_2_O and dried over Na_2_SO_4_. The solvent was evaporated under reduced pressure, and the residue was purified by flash column chromatography on Al_2_O_3_ (CH_2_Cl_2_: EtOAc = 30:1 (*v*/*v*)) to give **1aad** (24 mg, 0.05 mmol) as a reddish solid in 47% yield. Mp = 150 °C (decomp.). ^1^H NMR (400 MHz, CDCl_3_) δ 8.01–7.99 (m, 2H), 7.79–7.77 (m, 1H), 7.60–7.55 (m, 2H), 7.40–7.38 (m, 2H), 7.28–7.24 (m, 2H), 6.77–6.74 (m, 1H), 6.60–6.65 (m, 1H), 3.50–3.44 (m, 4H), 1.74–1.68 (m, 4H), 1.58–1.54 (m, 2H). ^13^C NMR (100 MHz, CDCl_3_) δ 159.1, 158.8, 158.5, 154.7, 154.5, 151.6, 151.6, 146.7, 146.7, 146.5, 133.1, 133.0, 131.8, 131.6, 129.0, 123.8, 123.7, 117.3, 117.3, 117.1, 117.0, 115.3, 113.8, 113.1, 110.5, 106.4, 98.9, 48.4, 25.5, 24.5. UV-vis (CH_2_Cl_2_, *c* = 1.1 × 10^–5^ M) λ_max_ (*ε*) 472 (20,600), 335 (19,200) nm. HRMS (ESI, positive mode): *m*/*z* calcd for C_32_H_21_NO_6_ [M]^+^ 515.1364; found: 515.1363.

**Synthesis of 1add.** To a 50 mL Schlenk tube was added **8add** (100 mg, 0.09 mmol), excess activated Cu Powder (1300 mg, 20.5 mmol), and dry DMF (28 mL) under argon atmosphere. The reaction mixture was refluxed for 18 h. After cooling to ambient temperature, the reaction mixture was filtered through Celite and washed with EtOAc. The organic phase was then washed with H_2_O and dried over Na_2_SO_4_. The solvent was evaporated under reduced pressure, and the residue was purified by flash column chromatography on Al_2_O_3_ (CH_2_Cl_2_: EtOAc = 30:1 (*v*/*v*)) to give **1add** (26 mg, 0.04 mmol) as a reddish solid in 47% yield. Mp = 115 °C (decomp.). ^1^H NMR (400 MHz, CDCl_3_) δ 8.98–7.96 (m, 1H), 7.78–7.76 (m, 2H), 7.55–7.51 (m, 1H), 7.37–7.35 (m, 1H), 7.25–7.21 (m, 1H), 6.76–6.72 (m, 2H), 6.65–6.64 (m, 2H), 3.49–3.37 (m, 8H), 1.77–1.64 (m, 12H). ^13^C NMR (100 MHz, CDCl_3_) δ 159.6, 159.3, 159.0, 154.5, 154.5, 154.3, 154.3, 151.5, 147.2, 146.9, 146.9, 132.6, 131.7, 131.7, 130.7, 123.5, 117.6, 116.8, 113.0, 112.2, 111.2, 110.4, 106.7, 106.6, 99.1, 99.0, 48.5, 25.5, 24.5. UV-vis (CH_2_Cl_2_, *c* = 1.1 × 10^–5^ M) λ_max_ (*ε*) 463 (44,900), 357 (14,400) nm. HRMS (ESI, positive mode): *m*/*z* calcd for C_37_H_30_N_2_O_6_: [M + H]^+^ 599.2177; found: 599.2176.

**Synthesis of 1ccd.** To a 50 mL Schlenk tube was added **8ccd** (100 mg, 0.09 mmol), excess activated Cu Powder (1330 mg, 20.5 mmol), and dry DMF (28 mL) under argon atmosphere. The reaction mixture was refluxed for 18 h. After cooling to ambient temperature, the reaction mixture was filtered through Celite and washed with EtOAc. The organic phase was then washed with H_2_O and dried over Na_2_SO_4_. The solvent was evaporated under reduced pressure, and the residue was purified by flash column chromatography on Al_2_O_3_ (CH_2_Cl_2_: EtOAc = 20:1 (*v*/*v*)) to give **1ccd** (36 mg, 0.06 mmol) as a reddish solid in 64% yield. Mp = 222 °C (decomp.). ^1^H NMR (400 MHz, CDCl_3_) δ 7.79–7.77 (m, 3H), 6.75–6.72 (m, 1H), 6.64–6.63 (m, 1H), 6.60–6.57 (m, 2H), 6.48–6.47 (m, 2H), 3.45–3.39 (m, 4H), 3.10 (s, 12H), 1.72–1.66 (m, 6H). ^13^C NMR (100 MHz, CDCl_3_) δ 159.9, 159.9, 159.9, 154.3, 154.2, 154.1, 153.6, 1477, 147.6, 147.4, 131.6, 110.3, 110.2, 110.1, 108.4, 107.1, 106.2, 99.2, 97.2, 77.5, 77.2, 76.8, 48.5, 40.2, 25.5, 24.5. UV-vis (CH_2_Cl_2_, *c* = 1.1 × 10^–5^ M) λ_max_ (*ε*) 453 (62,900) nm. HRMS (ESI, positive mode): *m*/*z* calcd for C_36_H_31_N_3_O_6_: [M]^+^ 601.2207; found: 601.2208.

**Synthesis of 1cdd.** 50 mL Schlenk tube was added **8cdd** (101 mg, 0.09 mmol), excess activated Cu Powder (1300 mg, 20.5 mmol), and dry DMF (27 mL) under argon atmosphere. The reaction mixture was refluxed for 18 h. After cooling to ambient temperature, the reaction mixture was filtered through Celite and washed with EtOAc. The organic phase was washed with H_2_O and dried over Na_2_SO_4_. The solvent was evaporated under reduced pressure, and the residue was purified by flash column chromatography on Al_2_O_3_ (CH_2_Cl_2_: EtOAc = 20:1 (*v*/*v*)) to give **1cdd** (28 mg, 0.04 mmol) as a reddish solid in 49% yield. Mp = 125 °C (decomp.). ^1^H NMR (400 MHz, CDCl_3_) δ 7.79–7.77 (m, 3H), 6.74–6.72 (m, 2H), 6.64–6.63 (m, 2H), 6.59–6.57 (m, 1H), 6.48–6.47 (m, 1H), 3.44–3.40 (m, 8H), 3.10 (s, 6H), 1.72–1.67 (m, 12H). ^13^C NMR (100 MHz, CDCl_3_) δ 159.9, 159.9, 159.8, 154.3, 154.2, 154.1, 153.6, 147.7, 147.4, 147.4, 131.6, 110.4, 110.4, 110.3, 110.2, 108.4, 107.1, 106.2, 99.2, 97.2, 48.5, 40.2, 25.5, 24.5. UV-vis (CH_2_Cl_2_, *c* = 1.1 × 10^–5^ M) λ_max_ (*ε*) 453 (55,400) nm. HRMS (ESI, positive mode): *m*/*z* calcd for C_39_H_36_N_3_O_6_: [M + H]^+^ 642.2599; found: 642.2598.

## 3. Results and Discussion

The synthetic route to *C*_3_ symmetrical compounds **1A**–**E** is depicted in Figure 1. The central benzene core **6** was efficiently prepared from mesitylene (**2**) in five steps, according to the literature procedures [32,33,34,35]. Compound **6** was reacted with the corresponding 2-bromophenol derivatives in the presence of K_2_CO_3_ in acetone to afford precursors **8A**–**E** in good yields. Finally, the targeted molecular triskelions **1A**–**E** were obtained via an intramolecular Ullmann coupling reaction. The *C*_3_ symmetric derivatives were fully characterized using ^1^H and ^13^C NMR spectroscopy and high-resolution mass spectrometry (HRMS) [28,29]. 

Subsequently, we employed the synthetic routes for **1A**–**E** (Figure 2) for the synthesis of asymmetric triskelions. First, **6** and 1.0 or 2.0 eq. of **7d** were treated with K_2_CO_3_ in acetone to give mono-substituted **9** and di-substituted **10** in 19% and 50% yield, respectively. Next, asymmetric precursors **8aad**, **8add**, **8ccd**, and **8cdd** were obtained in 52–74% yield using 2.0 eq. of **7a** or **7c** for **9**, and 1.0 eq. of **7a** or **7c** for **10**. Owing to the nucleophilic reactivity of **6**, it was difficult to introduce three different-type phenyl units into the central benzene core. The intramolecular Ullmann coupling of **8aad**, **8add**, **8ccd**, and **8cdd** in the presence of excess copper in DMF afforded the desired asymmetric triskelions **1aad**, **1add**, **1ccd**, and **1cdd**, which were analogous to **1A**–**E**. Although the ^1^H and ^13^C NMR spectra of these derivatives were highly complex due to their *C*_1_ symmetry, the asymmetric triskelions structures were identified (Appendix A). In this coupling condition, several debrominated compounds of **8**, mono- and bis-coupling products were observed as the byproducts, however, they could not be isolated.

The molecular structures of **1A**–**D** were confirmed using X-ray crystallography [28,29]. As expected, the molecular framework of **1** presented a helical geometry arising from intramolecular steric repulsion. This result was in good agreement with the corresponding optimized structures derived from quantum chemical calculations. Similarly, **1aad**, **1add**, **1ccd**, and **1cdd** were optimized as twisted propeller-shaped structures (Figure 2). Furthermore, in the case of these asymmetric triskelions, left-handed *M,M,M*- and right-handed *P,P,P*-formers were the most stable conformation regardless of the substituted groups.

The maximum transition state energy for the isomerization between (*P,P,P*) and (*M,M,M*)-**1A** was estimated to be 9.46 kcal/mol at the ωB97-XD/6-31G(d) level of theory. Even in the case of **1C**, bearing *N*-containing donor groups, the corresponding energy was 8.7 kcal/mol. This indicates that these derivatives underwent continuous flipping in diluted solutions. Furthermore, all triskelion-shaped compounds adopted a Janus-type structure and presented two distinct molecular π-surfaces (Figure 3). Based on the helical triskelion-shaped framework, the electrostatic potential mapping of these compounds demonstrated that the outer phenyl groups on the front surface of **1** were positively charged (blue) and the three electron-withdrawing carbonyl groups on the back face were negatively charged (red), regardless of the molecular symmetry. Thus, these molecules exhibit out-of-plane anisotropy via the central benzene ring from the positive area on the front to the negative area on the back face. The estimated dipole moments of the derivatives bearing donor groups reached 7.83 Debye, which is higher than those of bowl-shaped sumanene [36] and Janus-type subphthalocyanine [37]. Interestingly, the dipole moments of asymmetrical derivatives **1aad**, **1add**, **1ccd**, and **1cdd** were also showed large owing to the triskelion-shaped framework. We believe that this Janus-type anisotropy is beneficial for nanomaterials applicable in molecular recognition that operates through molecular rearrangement control [38].

The absorption and fluorescence spectra of asymmetrical triskelion-shaped coumarin dyes **1aad**, **1add**, **1ccd**, and **1cdd** were recorded in toluene to investigate their photophysical properties and compare them with those of *C*_3_ symmetrical dyes **1A**, **1C**, and **1D** (Figure 4). As shown in Figure 4a, in the absorption spectra of the symmetrical dyes, a single peak was observed at 342 nm for **1A** and at 443 nm for **1C** and **1D**. In contrast, two peaks were observed in the absorption spectra of **1aad** (335 and 472 nm) and **1add** (357 and 463 nm). On the other hand, the absorption spectra of **1ccd** and **1cdd** only contained one peak at 443 nm, similar to the results for the symmetrical dyes. These results indicated that the absorption maxima of **1aad** (λ_max_: 472 nm) and **1add** (λ_max_: 463 nm) exhibited a bathochromic shift compared to that of **1A** (λ_max_: 342 nm). This shift stems from the narrow gap between the highest occupied molecular orbital (HOMO)–lowest unoccupied molecular orbital (LUMO) resulting from reduced molecular symmetry (Figure 5). Conversely, the absorption bands of **1ccd** and **1cdd** at 443 nm resembled that of **1C** (λ_max_: 443 nm). Although there is not much difference among **1aad**, **1add**, **1ccd**, and **1cdd**, the HOMO and LUMO levels of **1ccd** and **1cdd** were similar to those of *N*-substituted *C*_3_ triskelions due to the presence of strong donor groups, despite their *C*_1_ molecular symmetry.

To further elucidate the photochemical behavior of the asymmetrical triskelion-shaped coumarin dyes, their excited states were examined using quantum chemical calculations. Figure 6 shows the absorption spectra simulated using quantum chemical calculations and the main components of the natural transition orbitals (NTOs) for each electronic excitation. The simulated absorption spectra are in good agreement with the experimental results (Figure 4a), wherein two peaks were observed in the spectra of **1aad** and **1add**, and a single peak for **1ccd** and **1cdd**. In the case of **1aad**, the NTO analysis shown in Figure 6 indicates that the S_0_ → S_2_ transition, the peak of which is observed in the longer-wavelength region, mainly involves the *N*-substituted moiety in the molecule (**d** unit), whereas the S_0_ → S_4_ transition, the peak of which appears in the shorter wavelength region, mainly involves two unsubstituted moieties (**a** units). Similarly, for **1add**, the S_0_ → S_3_ transition with a peak in the long-wavelength region involves two *N*-substituted moieties (**d** units), whereas the S_0_ → S_4_ transition with a peak in the short wavelength region involves one non-substituent moiety (**a** unit). On the other hand, in the cases of **1ccd** and **1cdd**, the S_0_ → S*_n_* (*n* = 2, 3, and 4) transitions all involve intramolecular *N*-substituted moieties, and their transition energies do not differ significantly; thus, they are observed as a single peak in the long wavelength region of the absorption spectra. Another interesting point here is that the oscillation intensity of S_0_ → S_4_ transition is smaller than that of S_0_ → S_2_ in the case of **1aad** and **1add**, but larger in the case of **1ccd** and **1cdd**. In general, the transition probability depends on the overlap between the molecular orbitals involved in the electronic excitation. Indeed, as shown in Figure 6, the overlap of the hole-particle pairs of NTOs for the S_0_ → S_4_ transition is small in the case of **1aad** and **1add**, but large in the cases of **1ccd** and **1cdd**. That is, due to the combination of the substitute positions, this framework was found to be designable for controlling the frontier orbitals and electronic transitions. In contrast, the NTO analysis of *C*_3_ symmetrical derivatives **1A**–**1E** suggested that their corresponding S_0_ → S_2_ or S_0_ → S_3_ transitions were composed of two pairs with large eigenvalues (Appendix A). This is due to the degenerated molecular orbitals resulting from their *C*_3_ symmetry. Therefore, distribution and overlapping from the hole to particle cover the whole of the molecular framework.

In the fluorescence spectra of **1aad**, **1add**, **1ccd**, and **1cdd** in toluene, the emission maxima were observed at 619, 606, 593, and 596 nm, respectively, with relatively large Stokes shifts of 5700–5800 cm^−1^ (Figure 4b). These emission maxima were as high as that of **1D** (599 nm) or slightly bathochromically shifted. The fluorescence quantum yields of these compounds were low (*Φ*_F_ = 0.003–0.007). This inefficiency is possibly caused by thermal non-radiative transitions arising from their helical inversion, in addition to the internal conversion from S_n_ (*n* = 2~4) to S_1_. These values are lower than those of obtained for the *C*_3_ symmetrical compounds **1C** (*Φ*_F_ = 0.14) and **1D** (*Φ*_F_ = 0.13) in toluene. The orbital degeneracy can be resolved by varying the substitution style on the molecular triskelion. As a result, the orbital overlap between the ground and excited states of the asymmetric derivatives would be reduced owing to their partial localization as shown in NTO analysis.

Next, we investigated the solvent dependence of the photophysical properties of **1aad**, **1add**, **1ccd**, and **1cdd** (Appendix A). All of the compounds demonstrated a gradual bathochromic shift in their absorption spectra depending on the solvent polarity. Conversely, the asymmetrical dyes, expect for **1ccd**, exhibited a blue-shift in their fluorescence spectra in highly polar solvents such as DMF and DMSO. The THF and EtOAc solutions of these compounds gave rise to broad emission bands containing shoulder peaks. This tendency was observed in the case of **1C** and **1D** [29]. This indicates that asymmetrical compounds undergo at least two radiative decay processes. Intriguingly, **1ccd** exhibited a redshift in highly polar solvents. For triskelion-shaped scaffolds, the charge separation structures in the excited states may be controllable depending on the combination of the three substituents. Based on the photophysical data, we evaluated the intramolecular charge transfer (ICT) characteristics of **1aad**, **1add**, **1ccd**, and **1cdd** by using Lippert-Mataga plots (Figure 7) [39,40]. Compounds **1aad**, **1add**, and **1cdd** exhibited a negative correlation, whereas **1ccd** showed a minimally increasing trend. However, linearity was poor in all cases, indicating that the ICT characteristics of these asymmetrical derivatives depend minimally on solvent polarity, in contract to typical coumarin dyes. It is considered that the resonance structures of **1** in the excited state are destabilized with the loss of aromaticity of the central benzene ring. The fluorescence quantum yields in the tested solvents were considerably low. The cause of low efficiency under diluted solutions is presumed to occur due to the dominant non-radiative process resulting from the helical inversion. We reasoned that fluorescence efficiency can be improved by suppression of molecular inversion in the aggregates, as in the case of AIE-gens [3].

To investigate the AIE enhancement (AIEE) properties of **1aad**, **1add**, **1ccd**, and **1cdd**, we first prepared THF/H_2_O solutions (water fraction (Fw): 10–80%) of **1aad**, **1add**, **1ccd**, and **1cdd**, and measured their absorption (Appendix A). For all the compounds, the absorption maxima gradually decreased as the Fw increased to 50–60%. At Fws of ≥60%, the absorbance decreased drastically, and scattering absorption bands were observed in the long-wavelength region. This result indicated that these compounds formed nanoaggregates in the THF/H_2_O mixture. Moreover, the DLS results indicated the existence of nanoaggregates with constant particle sizes in the range of 2~700 nm (Appendix A). As shown in Figure 8, the fluorescence spectral peaks of **1aad**, **1add**, **1ccd**, and **1cdd** at Fws of 10–50% were very low-intensity. Conversely, as the Fw increased to ≥60%, the intensity increased drastically owing to the formation of nanoaggregates. The maximum intensity was observed at Fw = 70%. This suggests that the AIEE characteristics of these asymmetrical triskelions arise due to molecular motions, such as vibrations and inversion, which are restricted in the nanoaggregates. Interestingly, these compounds exhibited a bathochromic shift of approximately 650 nm. Newly formed nanoaggregates of **1aad**, **1add**, **1ccd**, and **1cdd** may be affected by the reorientation of H_2_O in their excited state. However, at an Fw of 80%, the intensity decreased owing to reprecipitation. Thus, these asymmetrical derivatives are potentially applicable as components of effective AIE-gens owing to their twisted geometry, similar to *C*_3_ symmetrical triskelion-shaped dyes. The introduction of several substituents to the triskelion scaffold enabled the tuning of their orbital levels and molecular packing. It is interested in their emission behavior in the solid-state, such as powder, crystalline, and thin films.

## 4. Conclusions

We successfully synthesized Janus-type triskelion-shaped fluorophores comprising variably substituted coumarins (**1A**–**E**, **1aad**, **1add**, **1ccd**, and **1cdd**) via the copper-mediated intramolecular transannulation Ullmann reaction. Based on the optimized structures, we reasonably assumed that asymmetrical triskelions **1aad**, **1add**, **1ccd**, and **1cdd** would exhibit similar distorted structures of *C*_3_ symmetric triskelions **1A**–**1E**. All of the compounds presented out-of-plane anisotropy with large dipole moments (5.81–7.83 Debye) owing to the Janus-type structure, which presents two different π-surfaces. The absorption and emission maxima of the asymmetrical compounds **1aad**, **1add**, **1ccd**, and **1cdd** were redshifted compared to those of *C*_3_ symmetrical compounds **1A**, **1C**, and **1D**, owing to the narrow HOMO-LUMO gaps of the former. The emission behavior of the asymmetrical compounds, with the exception of **1ccd**, was weakly associated with solvent polarity. The quantum yields of **1aad**, **1add**, **1ccd**, and **1cdd** were considerably lower than those of the *C*_3_ symmetrical derivatives. NTO analysis suggested that the molecular orbitals of the *C*_3_ symmetrical compounds in the excited state are delocalized over the triskelion framework in the S_0_ → S*_n_* (*n* = 2, 3, and 4) electronic transition. Conversely, those of the asymmetrical compounds are located on one or two coumarin units through the central benzene ring. Therefore, we reasoned that the low emission efficiencies of **1aad**, **1add**, **1ccd**, and **1cdd** resulted from orbital distribution overlap. All the compounds formed nanoaggregates in THF/H_2_O mixtures and demonstrated AIEE characteristics arising from their distorted geometry. Thus, Janus-type triskelion-shaped AIE fluorophores are potentially promising candidates for the development of solid-state fluorescent and chiral materials, which can be optimized by controlling their molecular rearrangement in the solid state.

## Data Availability

Not applicable.

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
