# Peer review of "Janus-Type AIE Fluorophores: Synthesis and Properties of π-Extended Coumarin-Bearing Triskelions"

_molecules, 2022, doi:10.3390/molecules27217450_

Round 1

Reviewer 1 Report

The article titled: ‘Janus-type AIE Fluorophores: Synthesis and Properties of p-ex- 2

tended Coumarin-bearing Triskelions” concerns the fluorescence properties for 10 new synthesid derivatives. Authors also describe some electronic properties for old published structures (references 14,15). This causes the big missunderstanding.

Authors should very clearly separate and write it what has been already done (ref. 14, 15) and what is a subject of current research and publication.

Therefore it is need to rewrite introduction, and very clearly mention about it in the introduction.

Line 63,64 : herein we describe the synthesis, the structure…etc. In this publication the structure was not mention. But the discussion of structure and unit cell packing would be very interesting in context of fluotescence properties. If it is the object of ref 14,15 please exclude it from the article, if not please consider the more deeply discussion of the crystal structure. Within the respect of this please correct the experiemntal part.

Line 412, 413 “ racemic mixture of …” In the crystal structure description thereis no mentioned about it.

Reagrding the structures : ccdc1989929 (EMEBIK), 1964788 (EMEBOQ), 2108238 (DAYHEU), 2108239 (DAYHIY), please explain where the structures were published. 

Reviewer 2 Report

The paper can be accepted as it is.

Author Response

Thank you for sparing your valuable time.

Reviewer 3 Report

In this manuscript, Ueda and co-workers have reported the synthesis of differently substituted coumarin derivatives using the copper-mediated intramolecular trans-annulation Ullmann reaction to generate a number of symmetric and assymetric Janus-type triskelion-shaped fluorophores. The triskelion-shaped helically chiral geometry of synthesized C3 symmetric coumarin derivatives has successfully been proved through X-ray crystallographic analysis and through their appearance as a racemic mixture of left-handed (MMM form) and right-handed isomers (PPP form). Density functional theory (DFT) calculations also showed these molecules can remain in two rapidly flipping enantiomeric structures with a very low energy barrier and present two different molecular surfaces, similar to Janus-type molecules, and have considerably large dipole moment. The absorption and fluorescence spectra of asymmetrical derivatives exhibited a bathochromic shift compared to the symmetric coumarin derivatives due to their narrow HOMO–LUMO gap. Natural transition orbital analysis also showed difference in excited state orbital overlaps among the C3 symmetrical and asymmetrical dyes. They found that these triskelion-shaped fluorophores form molecular nanoaggregates in THF/H2O mixtures to demonstrate aggregation-induced emission (AIE) enhancement due to restricted molecular inversion. In view of these properties, they envisaged the potential applicability of these AIE fluorophores as solid-state chiral fluorescent materials with better tunability through alteration of substituents.

The synthesis of coumarin based chiral Janus-type organic fluorophores and the study of their photophysical properties and their corroboration through theoretical investigations are not very common. Many researchers working in the related fields will be interested in the results. Therefore, I recommend publication of this manuscript in Molecules.

Author Response

Thank you for sparing your valuable time. Your comments are a good encouragement for us.

Reviewer 4 Report

The manuscript entitled "Janus-type AIE Fluorophores: Synthesis and Properties of p-extended Coumarin-bearing Triskelions" by Ueda and co-woekers describes important findings and has strong scientific value. The bathocromic shift in the case of asymettric coumarin containing fluorophores in comparison to symmetrical fluorophores is really appreciable. The theoretical calculation using density functional theory also indicated that all the compounds presented two different molecular surfaces. Their investigation revealed that Janus-type aggregation induced emmision fluorophores are potentially useful as solid-state fluorescent chiral materials.

Although the manuscript has novelty in its findings, some points need to be taken care of before publication.

1. Page 3, line 82 and page 6, line 241,  HRMS means high resolution mass spectrometry, not spectroscopy.  Please correct it.

2. Page 6, Scheme-1, mentions the full name of BPO.

3. On page 6, Scheme 1, specify the reagent equivalent (Br2, Fe, AcONa, KMnO4, SOCl2, K2CO3, and so on).

4. Page 6, Scheme 1, conversion from compound 5 to 6, mentions the concentration of aqueous NaOH used.

5. Page 6, Scheme 2, mentions the reaction time and amount of K2CO3 used.

6. Explain briefly why the oscillator strength of So-S4 is less than that of So-S2 in the case of 1aad, whereas it is greater in the case of compound 1cdd.

7. Add some recent reviews on fluoropbore like, - (a) Journal of Pharmaceutical Analysis 2020, 10, 434-443.(b)  Current Chinese Chemistry 2022; 2(2):e020222200770 . https://dx.doi.org/10.2174/2666001602666220202142858  (c) Org. Biomol. Chem., 2021,19, 933-946.

8. Yield of the Ulmann coupling products are low to moderate, does the authors isolate any mono- or bis- coupling products along with the desired tris- coupling product?

Therefore, I am recommending this manuscript for acceptance after minor revision.

Round 2

Reviewer 1 Report

n/a